# Effects of Marine Microalgae on the Developmental Growth of the Sea Urchin Larviculture *Anthocidaris crassipina*

Yiru Chu [1,2], De-Sing Ding [1,*], Wei-Ting Sun [1], Cyril Glenn Satuito [2] and Chih-Hung Pan [1]

[1] Department and Graduate Institute of Aquaculture, National Kaohsiung University of Science and Technology, Kaohsiung 811, Taiwan; f107177106@nkust.edu.tw (Y.C.)

[2] Graduate School of Fisheries and Environmental Sciences, Nagasaki University, Nagasaki 852-8521, Japan; satuito@nagasaki-u.ac.jp

[*] Correspondence: 1041750102@nkust.edu.tw

**Abstract:** The sea urchin is a very important aquatic economic organism in many countries and has high food value. However, it has recently been heavily fished, and it would be of great importance to the sustainable development of coral reefs to develop large-scale aquaculture of sea urchins. Sea urchins are prone to death during larval development. Therefore, in this study, three kinds of microalgae were used as the initial food for sea urchins to evaluate whether different kinds of microalgae can improve their survival and growth rates. *Chaetoceros muelleri* (C), *Isochrysis galbana* tml (I), and *Dunaliella salina* (D) were fed to *A. crassipina* at concentrations of 5000, 10,000, 20,000, and 30,000 cell mL$^{-1}$. A fasted group was used as control (N). The final body length, final body width, final stomach length, rudiment length, survival rate and morphology were measured to evaluate development and growth. The results showed that feeding with *C. muelleri* resulted in better growth and survival. After 9 days of feeding with C(20,000 cells mL$^{-1}$), the rudiment length reached 203.33 ± 12.47 μm. The onset of metamorphosis was observed 12 days post-feeding. The survival rate after feeding C was also significantly higher than that after feeding I and D. In summary, when sea urchins are breeding, it is recommended to choose C(20,000 cells mL$^{-1}$) or C(30,000 cells mL$^{-1}$) as the initial feed for larvae to increase the growth and survival of sea urchin seedlings.

**Keywords:** *Anthocidaris crassipina*; *Chaetoceros muelleri*; development; *Dunaliella salina*; *Isochrysis galbana* tml; larval

**Key Contribution:** *C. muelleri* feeding can promote the smooth development of rudiments. *C. muelleri* feeding can promote the smooth metamorphosis of larvae.

## 1. Introduction

Sea urchin gonads are delicious and contain many proteins, amino acids, unsaturated fatty acids and other physiologically active substances. In recent years, many countries have regarded sea urchin gonads as a special food culture, and this has resulted in the development of important sea urchin markets in Taiwan, Japan and Europe [1]. In Taiwan, *Anthocidaris crassipina* is an important seafood. From May to September every year, a large number of purple sea urchins are caught and sold as seafood. Therefore, sea urchins have significant aquaculture economic value [2]. If large-scale production can be achieved through captivity, the sustainable development of the sea urchin aquaculture industry will be promoted. *Anthocidaris crassipina* is distributed in China, Japan, Taiwan, etc. In Taiwan waters, during the yearly breeding season from May to September, fishermen catch mature sea urchins for sale, resulting in a gradual decrease in production in recent years. According to the Sustainable Development Goals, SDG goal 14, Life Below Water, aims to practice conservation and sustainable use of oceans, seas and marine resources for sustainable development [3]. To solve the demand for sea urchins in the seafood market and for sea

urchin resources, large-scale aquaculture of sea urchins is crucial. If effective large-scale aquaculture can be achieved, the fishing of wild sea urchins will be reduced along with overfishing, and the ecological sustainability of coral reefs will be promoted.

Sea urchins are dominant benthic herbivores with important biological functions, such as removal of algae from coral reefs [4,5]. Sea urchins also have the ability to maintain the ecological sustainability of coral reefs. In recent years, environmental changes, ocean acidification and ultraviolet rays have caused serious threats to the embryonic development of sea urchins [6,7]. Therefore, large-scale aquaculture of sea urchins is imperative. The life cycle of sea urchins includes two stages, larval planktonic and adult benthic, with a high mortality rate in the larval stage. It is very important to improve the larval development, metamorphosis, growth and survival of sea urchins, which will in turn be beneficial to the large-scale aquaculture of sea urchins. Previous studies have found that *A. crassispina* grows in the sea where two algae, *Sargassum* spp. or *Corallina* spp., are distributed. Studies have noted that *Sargassum* spp. can promote growth and reproduction [4]. Ab et al. (2004) found that periphytic diatoms can induce *Pseudocentrotus depressus* larval metamorphosis in large-scale aquaculture. The consumption of different algae results in variations in the growth and development. In addition to sustaining sea urchins, algae also promote the metamorphosis of sea urchin larvae [8]. Sea urchins play an important role in coastal ecosystems, and their life cycle includes two stages: larval planktonic and adult benthic. It can feed on phytoplankton in the planktonic stage, and mainly feeds on macroalgae in the benthic stage. Sea urchin larvae mainly feed on microalgae, which can prevent these macroalgae from growing profusely, covering coral reefs and causing coral death [9].

*Anthocidaris crassipina* often die in large numbers during larviculture. If we can effectively explore what might constitute a proper initial food, it will be of great help to the large-scale aquaculture of sea urchins. The choice of adequate feed is very important as food nutrition has a strong relationship with embryonic development and metamorphosis [9]. According to research, sea urchin larvae mainly feed on microalgae, and choosing suitable algae for feeding will help the development of sea urchin larvae in captivity [10]. In this study, we explored the optimization of different microalgae as *A. crassipina* larvae stage feed. Suitable food shortens the time needed for sea urchin larval stage metamorphosis, thereby accelerating metamorphosis and settlement in order to achieve large-scale aquaculture of *A. crassipina*.

## 2. Materials and Methods

### 2.1. Sea Urchin Samples

Adult *A. crassipina* were obtained from the Gongliao sea urchin farm in northeast Taiwan. The reproductive season of this species of sea urchin in Taiwan is from May to September. The adult sea urchins were placed in a 120 × 40 × 20 cm glass tank with a photoperiod of 16 L: 8D at 26 °C. The water quality conditions are summarized in Table 1. In June 2022, KCl solution (0.5 M; 1 mL per individual) was used to chemically induce spawning [11]; a syringe was used to inject KCl solution into the peri-stomial membrane, and a total of 10 sea urchins were injected. The average weight of the adult sea urchins was approximately 141.049 ± 7.36 g. After the injection, the sea urchin was turned upside down and fixed above the 1000 mL beaker with the anus facing down (Figure 1) to facilitate the collection of sperm and eggs. Only the cloaca was soaked in seawater, and the oviposition and semen production of the sea urchin were observed.

**Table 1.** Sea urchins' culture water quality conditions.

| Water Quality Conditions | |
|---|---|
| Temperature | 26.00 ± 0.5 °C |
| pH | 8.0 ± 0.5 |
| Dissolved oxygen | 5.00 ± 0.05 ppm |
| Nitrous acid | 0.01 ± 0.05 ppm |
| Nitric acid | 0.05 ± 0.05 ppm |
| Calcium | 425 ± 30.12 ppm |
| Magnesium | 1345 ± 40.25 ppm |
| Ammonia nitrogen | 0.01 ± 0.05 ppm |
| Phosphate | 0.01 ± 0.01 ppm |

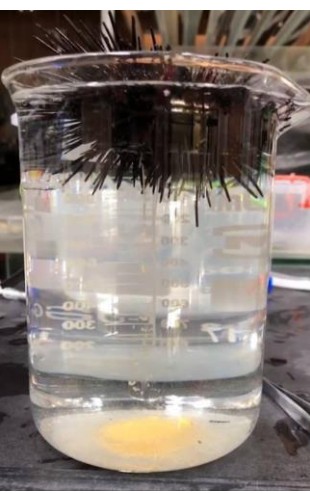

**Figure 1.** Adult *Anthocidaris crassipina* are releasing eggs.

### 2.2. Insemination and Hatchery

In this study, referring to the method of Castilla-Gavilán et al. 2018 [11], when male sea urchins were observed to discharge sperm, it was collected with a dropper and stored at 4 °C. After the female sea urchin laid her eggs, they were washed twice with sterilized seawater. First, a microscope (Leica DM500 microscope at 400×) was used to observe the shape and maturity of eggs and sperm and to determine them as such. A 1000 mL beaker filled with sterilized seawater was used to inseminate with a sperm/egg ratio of 1:4. After fertilization, the fertilized eggs were poured into plankton nets (500 mesh) and rinsed with sterile seawater. The fertilized eggs were rinsed at least three times to flush out excess sperm and prevent polyspermy and then stocked in a 60 × 30 × 20 cm glass tank. To prevent the fertilized eggs from sinking to the bottom, an air pump was used to inject air into the glass tank to disturb the water flow and increase the dissolved oxygen. The water quality conditions are summarized in Table 1. Every 30 min after fertilization, a microscope was used to observe the development of fertilized eggs and embryos, and pictures were taken to record the embryo development process.

### 2.3. Embryonic and Larval Development

Using a larval culture reference [11,12], 60 × 30 × 20 cm glass tanks were used for breeding, and 1/3 of the total water was changed every morning and evening with a larvae density of 4 individuals mL$^{-1}$. Fourteen hours after insemination, the larvae developed to blastulae; 20 h to gastrula; 38 h to prism stage; 46 h to 2-arm pluteus; and 48 h to 4-arm pluteus. In this study, sea urchins were fed 48 h after hatching according to the experimentally designed daily number of cells.

### 2.4. Microalgal Diets

For the cultivation method of microalgae, please refer to [13]. Three microalgae, *Chaetoceros muelleri* (C), *Dunaliella salina* (D) and *Isochrysis galbana* tml (I), were cultured in 2000 mL glass Erlenmeyer flasks containing Walne culture medium. High-temperature and high-pressure sterilization (121 °C, 30 min) was used before culturing. The culture was then cooled to 26 °C and irradiated using a fluorescent lamp with a 12 L:12 D photoperiod. After culturing for one week [13], the hemocytometer was used to calculate the number of microalgae.

### 2.5. Experimental Grouping

Forty-eight hours after insemination, sea urchin larvae were stocked in a 3000 mL transparent plastic tank. Each treatment group contained triplicates (with 4 individuals mL$^{-1}$ each) and a total of 12,000 individuals. In this experiment, three kinds of microalgae, C,

D, and I, were used as the initial bait for sea urchins. Feeding began 48 h after hatching. The feeding amounts were 5000, 10,000, 20,000 and 30,000 cell mL$^{-1}$. The group without feeding served as the control group (N). We observed the development of sea urchin larvae and the experiment continued until rudiment appeared. During the experiment, the final body length, final body width, final stomach length, post-oral arm length, rudiment length, final rudiment length, and survival rate were measured. Ten sea urchin larvae were taken from each group for measurement and analysis, and a Canon EOS 750D camera was used to record the shape of this larvae. The formula for calculating the survival rate is as follows:

$$\text{Survival \%} = \frac{\text{larvae in the rearing tanks}}{\text{larvae initially dispensed into the tanks}} \times 100$$

*2.6. Body Length Measurement and Morphology*

During this experiment, a microscope was used to measure daily body length, body width, post-oral arm length, stomach length, and rudiment length, as shown in Figure 2. In addition to measuring body length, the developmental morphology of larvae at different stages was also photographed and recorded. This experiment recorded the embryonic development, emergence of sessile pedicellariae, metamorphosis and juvenile development.

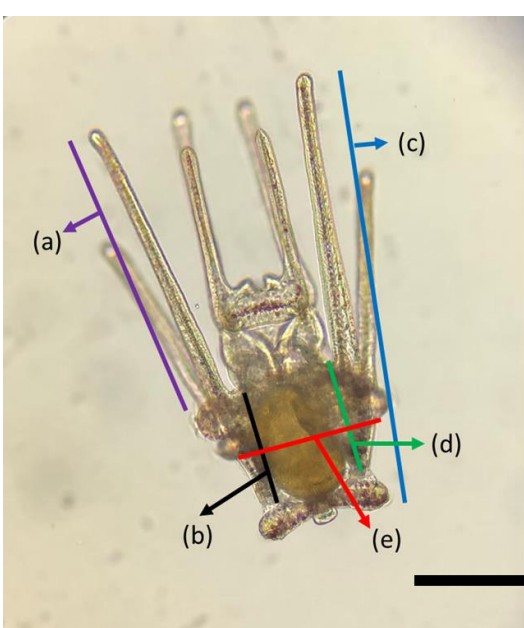

**Figure 2.** Measurement of *Anthocidaris crassipina* larvae. The scale bar is 200 μm. (**a**) post-oral arm length; (**b**) stomach length; (**c**) final body Length; (**d**) rudiment length; (**e**) final body width.

*2.7. Statistical Analysis*

After the experiment, the body length, body width, stomach length, and rudiment length measured on days 3, 6, and 9 of feeding were used for statistical analysis to check the growth and development rate. The results are expressed as the mean ± standard deviation. Statistical significance for growth, development, and survival was determined using one-way ANOVA with Duncan's multiple range test. A $p < 0.05$ was considered significant. All statistical analyses were performed using IBM SPSS version 20. These analyses were used to assess the growth and survival rates of sea urchin larvae on different diets.

## 3. Results

*3.1. Body Length and Body Width*

According to the experimental results, after 9 days of cultivation, the final body length of A. crassipina larvae was better in the C group (Figure 3); C(5000) 856.67 ± 12.47, C(10,000) 883.33 ± 12.41 μm, C(10,000) 883.33 ± 12.41 μm, C(20,000) 900 ± 16.33 μm,

C(30,000) 896.67 ± 26.25 µm, which was significantly improved compared with other groups ($p < 0.05$). Compared with the D group, the C group was significantly increased by 1.69, 1.72, 1.85 and 1.94 times, respectively. The C group increased by 1.78, 1.84, 1.87 and 1.87 times, respectively, compared with the control group. According to the results, feeding C can increase the body length of A. crassipina larvae. The results of final body width showed that the best treatment group was C, whose results were significantly different from those of the other treatment groups ($p < 0.05$). According to Figure 3, the final body width of D(30,000) is the shortest at 110 ± 8.16 µm; the longest is C(30,000) at 273.33 ± 12.47 µm, with a total difference of 2.48 times. C(30,000) increased by 2.05 times compared with N.

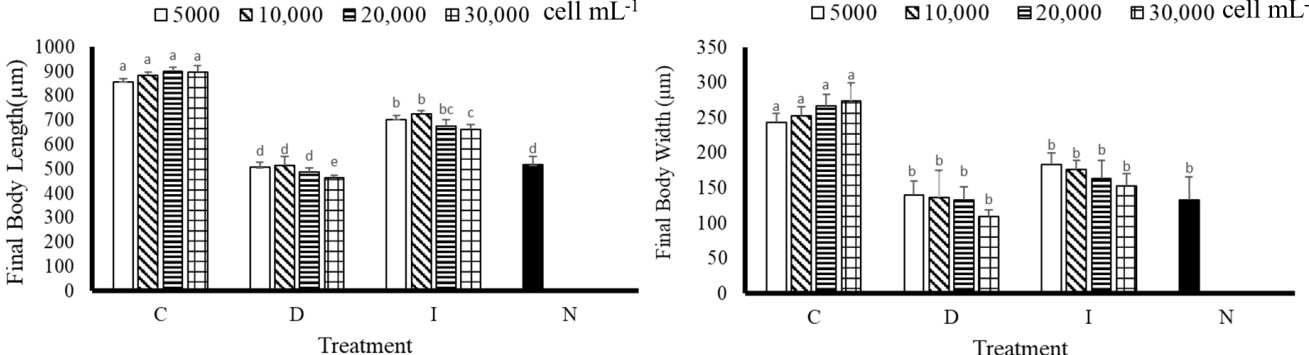

**Figure 3.** Final body length and final body width of sea urchin larvae after feeding for nine days. Bars represents ±SD (*n* = 3). The superscript English letters represent significant differences ($p < 0.05$). C: *Chaetoceros muelleri*, D: *Dunaliella salina*, I: *Isochrysis galbana* tml, N: control group.

### 3.2. Postoral Arm Length and Stomach Length

According to Figure 4, A. crassipina larvae fed with C had a better postoral arm length, with an average length of 793.33 ± 17.00 µm, and the shortest postoral arm length was D(30,000) 353.33 ± 4.71 µm, reflecting a difference of 2.25 times. That of the control group N was 413.33 ± 17.00 µm. The results showed that postoral arm length growth could be promoted by C feeding. It can also be observed that the C group had better stomach growth. Although there was no significant difference among the concentrations, it can be observed that the length of the stomach increased faster as the feeding amount increased. Except for the C group, there was no significant difference among the other treatment groups and the control group; in some cases, the higher the feeding concentration, the slower the growth of stomach length.

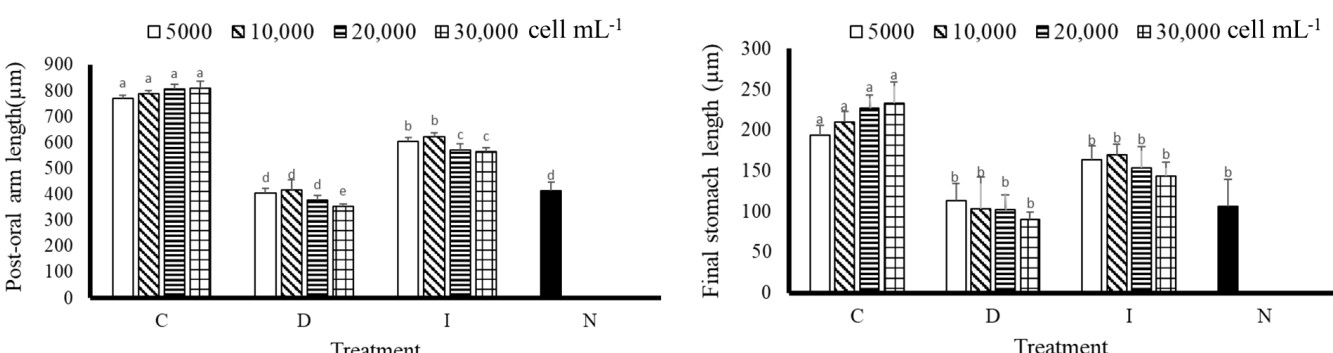

**Figure 4.** Post-oral arm length and final stomach length of sea urchin larvae after nine days of feeding. Bars represents ±SD (*n* = 3). The superscript English letters represent significant differences ($p < 0.05$). C: *Chaetoceros muelleri*, D: *Dunaliella salina*, I: *Isochrysis galbana* tml, N: control group.

### 3.3. Rudiment and Metamorphosis

Sessile pedicellariae appear on the top of the larvae that have developed to this stage. The sea urchin seedlings in this period are ready for metamorphosis; at this time, swimming and vitality will be weakened. It can be observed that the rudiment next to the stomach gradually increases over time (Figure 5H). According to Table 2, only the C group developed a rudiment, and this was not observed in the other groups. The rudiment length of C(30,000) was 203.33 $\pm$ 12.47 µm, C(20,000) 193.33 $\pm$ 4.71 µm, C(10,000) 173.33 $\pm$ 9.43 µm and C(5000) 153.33 $\pm$ 12.47 µm. The rudiment lengths of C(30,000) and C(20,000) were significantly longer than those of the other groups. Anthocidaris crassipina larvae that have developed to this stage gradually reduce swimming activity and sink to the bottom to begin metamorphosis.

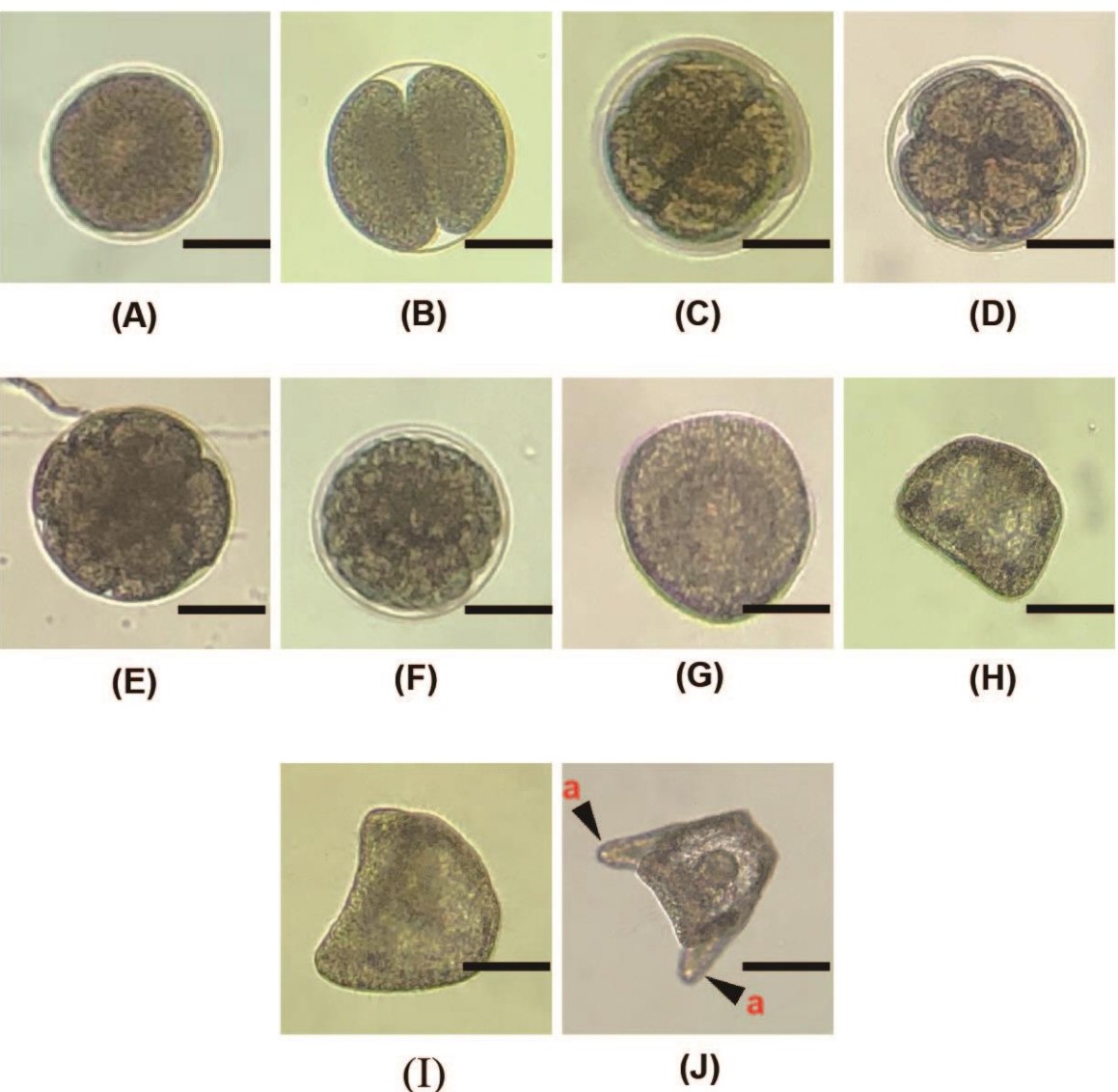

**Figure 5.** Morphology of *A. crassipina* embryonic development. The scale bar is 50 µm. (**A**) Fertilized egg, 30 min after insemination; (**B**) 2-cell stage, 1–2 h after insemination; (**C**) 4-cell stage; (**D**) 8-cell stage; (**E**) 16-cell stage, 4–5 h after insemination; (**F**) 32-cell stage; (**G**) morulla stage, 8–10 h after insemination; (**H**) the ciliated gastrula is formed 15 h after insemination. Larvae that have developed to this stage have already begun to swim. Floating on top of the water; (**I**) Prism; (**J**) 2-arm pluteus, 34 h after insemination; the mouth and digestive tract have not yet differentiated. a: The arm that has just started to grow.

**Table 2.** The final rudiment length and final survival rate of sea urchin larvae after feeding for 12 days.

| | Treatment | | | | | | | | | | | | |
|---|---|---|---|---|---|---|---|---|---|---|---|---|---|
| | 5000 Cell mL$^{-1}$ | | | 10,000 Cell mL$^{-1}$ | | | 20,000 Cell mL$^{-1}$ | | | 30,000 Cell mL$^{-1}$ | | | N |
| | C | D | I | C | D | I | C | D | I | C | D | I | |
| Final rudiment length (μm) | 153.33 [b] (12.47) | 0.00 (0.00) | 0.00 (0.00) | 173.33 [b] (9.43) | 0.00 (0.00) | 0.00 (0.00) | 193.33 [a] (4.71) | 0.00 (0.00) | 0.00 (0.00) | 203.33 [a] (12.47) | 0.00 (0.00) | 0.00 (0.00) | 0.00 (0.00) |
| Final survival Rate (%) | 21.67 [b] (2.36) | 3.23 [c] (0.17) | 1.67 [d] (0.18) | 25.00 [ab] (4.08) | 2.63 [c] (0.19) | 2.00 [d] (0.16) | 40.00 [a] (4.08) | 0.00 (0.00) | 0.00 (0.00) | 31.67 [a] (3.12) | 0.00 (0.00) | 0.00 (0.00) | 0.00 (0.00) |

The superscript to the right of values indicate horizontal comparisons and shows significant differences ($p < 0.05$). SD in ± ($n = 3$). C: Chaetoceros muelleri, D: Dunaliella salina, I: Isochrysis galbana tml, N: control group.

### 3.4. Survival

The survival rate of A. crassipina larvae that developed to the early juvenile stage is shown in Table 2. The highest survival rate was in the C group, and the survival rates from high to low feeding concentrations were 31.67 ± 3.12%, 40.00 ± 4.08%, 25 ± 4.08% and 21.67 ± 2.36%. According to the results, the higher the C feeding concentration, the higher the survival rate. In addition, the survival rates in D(5000) and D(10,000) were 3.23 ± 0.17% and 2.63 ± 0.19%, respectively. The survival rates of I(5000) and I(10,000) were 1.67 ± 0.18% and 2.00 ± 0.16%, respectively. D and I were fed 20,000 or 30,000 cell mL$^{-1}$, which was consistent with the control Group N, and the survival rate was 0%. Anthocidaris crassipina larvae can survive for 9 days without feeding, and the remaining survival rate is 2.17 ± 0.24% on the 6th day. Through observation, it is found that it only develops to 4-arm. In summary, A. crassipina larvae fed D and I had a low survival rate.

### 3.5. Developmental Morphology

#### 3.5.1. Cleavages and Blastula

After insemination, the fertilized egg begins to undergo cell division, as shown in Figure 5. The two-cell stage develops 1–2 h after insemination. If artificial insemination is carried out and egg cell division has not been found after 2 h, it may be due to incomplete fertilization or polyspermy that the fertilized eggs cannot successfully develop into embryos. Eight hours after insemination, the morulla stage develops, and the ciliated gastrula form after 15 h, at which time they can start to swim slowly. Thirty-four hours after insemination, the 2-arm pluteus develops. During this period, the mouth and digestive tract have not yet formed, so they cannot eat.

#### 3.5.2. Larval and Early Juvenile

The effects of C diets on the development of A. crassipina larvae are shown in Figure 6. In this experiment, 48 h after insemination, the feeding experiment started when the A. crassipina larvae had grown to 4-arm. According to the C group, with the best growth, it can be observed that, after 6 days of feeding, the larvae developed to 6-arm. After the sixth day, the seventh and 8-arm developed simultaneously. It can be observed that the arm development of sea urchin larvae is hyperplasia in pairs. On the eighth day after feeding, the rudiment began to appear next to the stomach, and gradually grew and increased over time. Until the 12th day after feeding, sessile pedicellariae appeared on the top of the larval stage; at this time, the sea urchin was ready for metamorphosis. This study found that if the rudiment was not mature enough, it would not be able to metamorphose smoothly, leading to death. The activity of A. crassipina larvae slows down at this stage, the arms arrange in parallel and the larvae settle to the bottom of the tank.

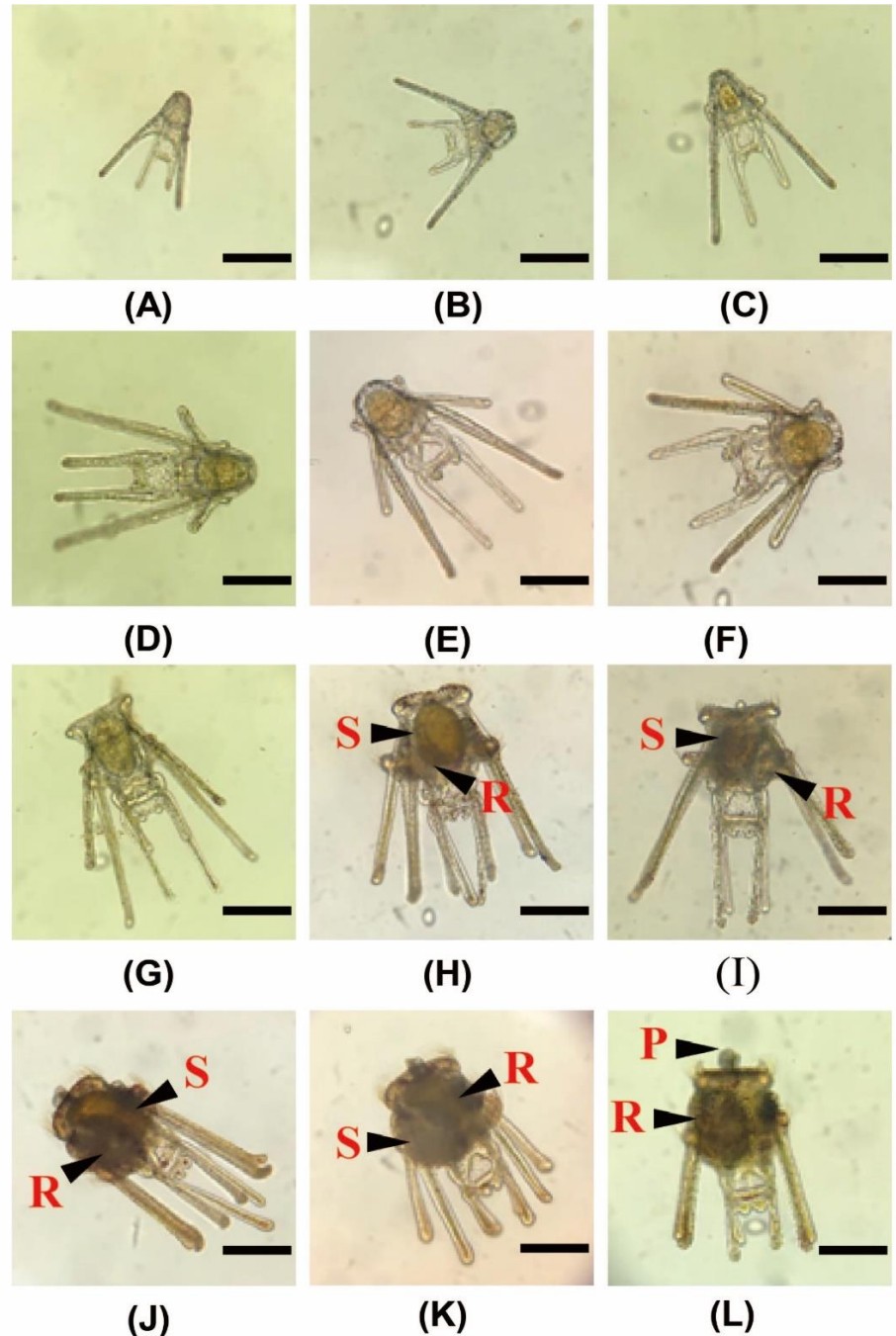

**Figure 6.** Morphology of *A. crassipina* larval development. The scale bar is 200 μm. (**A**) Prism: on the first day of feeding, 48.00 h after fertilization, the 4-arm pluteus larva has developed. It has a complete digestive tract and can start to eat microalgae. The hollow is in the middle is the mouth, and the arm will gather microalgae to the mouth as it swims to feed. (**B**,**C**) Feeding on the 3rd day: maintain 4-arm, but the arm gradually becomes longer. (**D**–**F**) After the 6th day of feeding: the 6-arm pluteus larva has developed, the postoral arm is elongated, and the postoral and anterolateral arms are also well developed. The digestive tract is located in the middle, and the bulbous stomach (S) can be clearly observed. (**G**–**I**) After feeding for 9 days, development occurred into 8-arm pluteus larvae. Rudiment (R), bulbous stomach (S). (**J**–**L**) Twelve days after feeding, the *A. crassipina* larvae have begun to prepare for metamorphosis, and it can be clearly observed that the rudiment gradually becomes larger. The wrists are arranged downward and parallel to each other. Sessile pedicellariae (P) appear.

Figure 7 shows the body shape and development of A. crassipina larvae in the three feeding treatments. On the third day after feeding, the morphology of the C and I groups were similar in the development of the arms, and the body lengths were 516.67 ± 12.47 μm and 606.67 ± 23.57 μm, respectively, while the arms of the D group were 443.33 ± 17.02 μm. On the 6th day after feeding, the lengths of the C, I, and D group arms were 743.33 ± 17.00 μm, 676.67 ± 26.25 μm, and 513.33 ± 38.59 μm, respectively. It can be observed that the arm length of the C group is longer than that of groups I and D. On the sixth day, it can be observed that the C and I groups developed to six arms. After 9 days of culture, the D group only developed to the eight-arm, and there was a case of stagnant development. Sessile pedicellariae were observed in the C group on Day 9, and rudiment could be observed in the body cavity. Sessile pedicellariae and rudiment were not observed in the I and D groups on Day 9. The D group began to exhibit growth arrest beginning on the sixth day. In summary, feeding C will effectively promote the developmental rate and metamorphosis of sea urchins. According to our observations, after 12 days of feeding, the larvae will begin to metamorphose into the juvenile phase. At this time, the shape is similar to that of adult sea urchins, with dult spines and extended tube-feet (Figure 8). In this study, it was only found that the bait can cause growth and development retardation and affect the rate of metamorphosis, and no deformity was found.

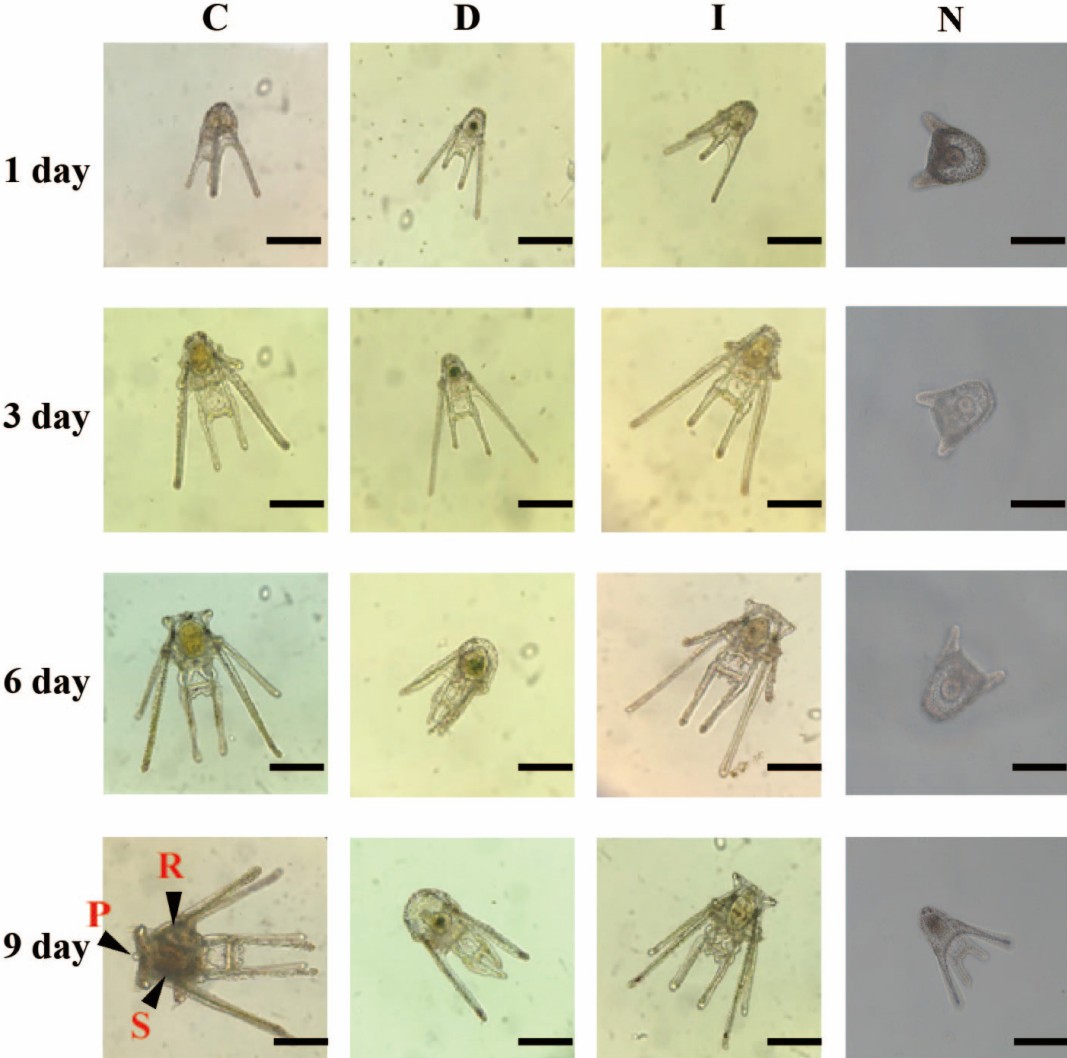

**Figure 7.** Differences in the developmental morphology of *A. crassipina* larvae fed different microalgae. rudiment (R), bulbous stomach (S), Sessile pedicellariae (P). The scale bar is 200 μm. C: *Chaetoceros muelleri*, D: *Dunaliella salina*, I: *Isochrysis galbana* tml, N: control group.

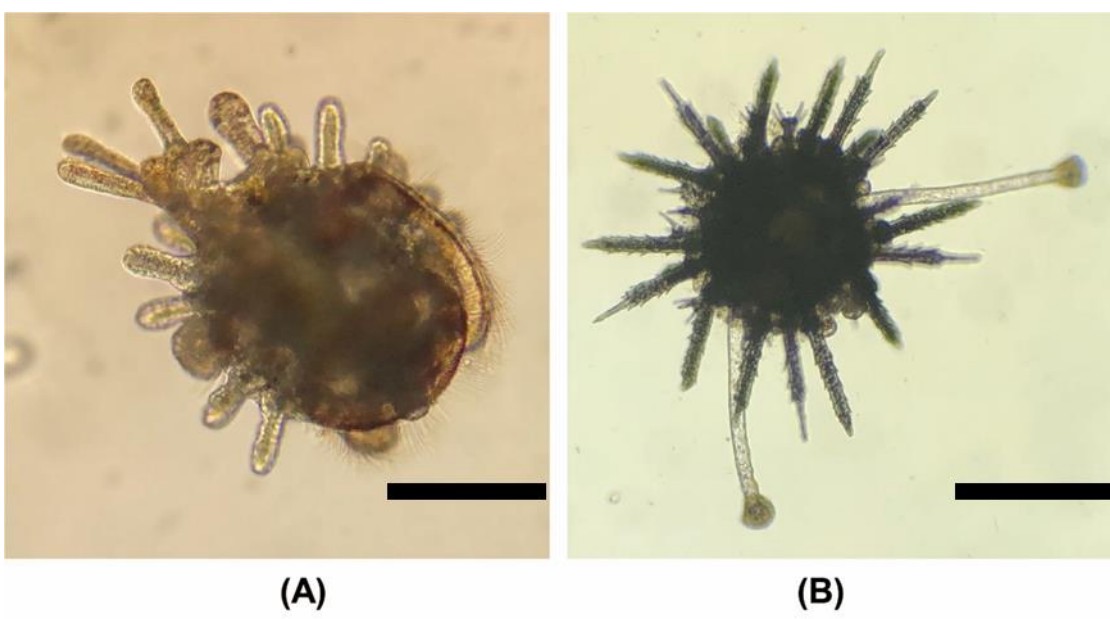

**Figure 8.** Larval metamorphosis and early juvenile stages. The scale bar is 200 μm. (**A**) Competent larvae with complete rudiment growth. Tube-feet protrude and fix on the tank surface and the larva gradually undergoes metamorphosis. (**B**) Juvenile, adult spines and extended tube-feet can be observed. The larva has completely sunk to the bottom at this time, and can only move with tube-feet.

## 4. Discussion

### 4.1. Growth and Survival

In this study, after feeding for 12 days, it was found that C can improve the larval development and metamorphosis of sea urchins. Previous studies have noted factors that affect the survival and growth of sea urchin larvae, including food ration, diet type, stocking density, temperature, water chemistry, salinity, dissolved oxygen and water chemistry [10]. This study found that feeding both D and I led to growth and larval development retardation, and the developmental delay was found 3 days after feeding. Developmental malformation was more severe due to nutritional deficiencies after 6 days of feeding. A previous study found that the survival rate of *Paracentrotus lividus* larvae was 1.19 times higher when fed *Chaetoceros gracilis* than when fed *Isochrysis* sp. [9]. Therefore, choosing the correct bait is beneficial to the development and survival of sea urchin larvae. This study is only a preliminary study of the effects of different algae on the growth and survival of *A. crassipina* larvae. According to the survival, larval growth and metamorphosis rates observed in the experiment, it is best to feed *C. muelleri*.

In addition, Kalam et al. (2010) [10] found that sea urchin larvae often use *Dunaliella tertiolecta* or *Chaetoceros* spp. as the initial bait; the feeding amount was 3000–9000 cells mL$^{-1}$ and 20,000–60,000 cells mL$^{-1}$, and different larval stages and stocking densities may affect the feeding amount. This study found that feeding *A. crassipina* larvae *C. muelleri* resulted in better survival and growth rates and is helpful for metamorphosis. Feeding *D. salina* and *I. galbana* tml can cause larval development to be delayed, and sea urchin seedlings fed *D. salina* only develop to the 4-arm stage before development ceases. Therefore, different microalgae diets may have a great influence on the larval development of different species of sea urchins. Previous studies have found that algae commonly used as bait for larval sea urchins include *C. gracilis*, *D. tertiolecta*, *Isochrysis galbana*, and *Rhodomonas* spp., and the feeding amount varies depending on the type of sea urchin. For example, Strongylocentrotus purpuratus is fed with *D. tertiolecta* at a feeding amount of 6000 cell mL$^{-1}$ [10,14]. In addition, *Tripneustes gratilla* was fed *C. muelleri*, and the feeding amount was 4000–10,000 cell mL$^{-1}$ [15]. *Strongylocentrotus franciscanus* uses *C. gracilis* 18,000 cell mL$^{-1}$ and *Rhodomonas lens* 7000 cell mL$^{-1}$ for mixed feeding [14]. There has



been no previous research on suitable bait for feeding *A. crassispina* larvae. This study found that *A. crassispina* larvae have better growth and survival rates when fed *C. muelleri* at 20,000–30,000 cell mL$^{-1}$, which can be applied to large-scale *A. crassispina* aquaculture.

According to previous studies, the stocking amount of sea urchins affects not only the amount of feeding but also the growth and survival of sea urchins [10]. In terms of stocking density, the sea urchin larvae density of four individuals mL$^{-1}$ was used in this study, which is suitable for breeding *A. crassispina* larvae. Other studies have shown that sea urchin larvae can survive and grow more effectively when the average stocking density is 1–2 individuals mL$^{-1}$, which is different from our findings [10]. The reason for this may be that different types of sea urchins have different suitable stocking densities. Other studies have noted that the *S. purpuratus* stocking density is 5–10 individuals mL$^{-1}$ [16]. The stocking density of Strongylocentrotus droebachiensis is 1 individual mL$^{-1}$ [17]. The stocking densities of the abovementioned studies were all different from the stocking densities of *A. crassispina* in this study. Therefore, it is necessary to choose a suitable stocking density when cultivating different types of sea urchin larvae.

*4.2. Larval Development*

Sea urchin fertilized eggs will sink to the bottom before they have developed into gastrula. An air pump should be used to generate water flow so as not to cause the embryos to pile up on each other at the bottom of the tank and cause death. We initially found that this may be a phenomenon caused by hypoxia or extrusion.

When the gastrula develops, it has the ability to float on the water. This study found that feeding *C. muelleri* significantly improved the arm development of *A. crassispina* larvae. However, feeding *D. salina* and *I. galbana* tml resulted in developmental stagnation, and the developmental retardation was the most serious on the sixth day. It can be observed that the arms of *A. crassispina* are all tightened or the arms overlap, affecting the float and movement of larvae. Although feeding *I. galbana* tml had no significant effect on the development of the wrist, no rudiment was found after 9 days of culture. *A. crassispina* begin to die in large numbers on the 10th day, which means that these sea urchins cannot metamorphose into juveniles smoothly. In the development stage of larvae, careful selection of suitable bait will be of great help to the large-scale aquaculture of sea urchins.

*4.3. Morphology*

From the observation of the juvenile development of *A. crassispina* larvae, it was found that the mouth and digestive tract were not formed until 48 h after hatching. When larvae develop into 4-arm pluteus larvae, they start to feed on microalgae. It has also been observed in *S. sphaeroides* that the larvae need to develop to the 4-arm stage to start feeding [18]. Therefore, 4-arm pluteus larvae can be used as the basis for judging the initial feeding time of *A. crassispina* larvae. As the sea urchin grows, it can develop into a 6-arm pluteus larva in 10 days. At this time, it is very active and will move up and down in the tank, so strong water flow should be avoided. Excessive water flow will easily cause the arms between sea urchin larvae to become stuck and sink, causing a large number of sea urchin seedlings to die. After 12 days of feeding, *A. crassispina* larvae will start metamorphosis. Before metamorphosis, the 8-arm appeared parallel and straight, and tentacles with suckers were found in the rudiment (Figure 8A). At this time, metamorphosis is not yet complete. A similar situation was also observed in the development of *S. sphaeroides* larva [18]. There is a high mortality rate during this developmental period. Previous studies have found that, without induction, larvae will not be able to successfully undergo metamorphosis [18]. Some studies have proposed a method of induction. Coralline red algae can be added to the tank to promote metamorphosis [8]. However, this study did not use this method. We only used *C. muelleri* for continuous feeding, but it can still promote the metamorphosis of *A. crassispina* larvae. It is judged that this may be a long-term cultivation process, resulting in the growth of sessile algae on the tank wall, and the substances released by these algae may promote sea urchin metamorphosis. Some scholars have previously proposed a

method to induce the metamorphosis of sea urchins, which can be mixed with coralline red algal extracts + C. diatom (50 : 50) and added to the tank. This mixture can promote the metamorphosis of *Salmacis sphaeroides* [18]. Therefore, algae or algae extracts may be the main promoting factor. At present, there are very few studies on the inducers of sea urchin metamorphosis, and further research should be performed in the future.

This study found that larval metamorphosis could be observed 12 days after feeding C. This is different from previous studies, which found that *S. sphaeroides* did not undergo larval metamorphosis until 26 days after culture [18]. Sea urchins will start to crawl after metamorphosis (Figure 8B). Large-scale seaweed or *Brassica oleracea* var. capitata can be provided as the initial bait for large-scale aquaculture, and attention should be given to changes in water quality to increase the breeding rate.

## 5. Conclusions

This study found that feeding *C. muelleri* can effectively improve the growth and survival rate of *A. crassispina* larvae and can promote metamorphosis. The study found that feeding *C. muelleri* 20,000–30,000 cell mL$^{-1}$ can promote the development of the *A. crassispina* larval arm and start metamorphosis 12 days after feeding. 34 h after insemination, it will develop to the 2-arm pluteus stage. During this period, the mouth and digestive tract have not yet formed, so it is not possible to eat. In artificial large-scale aquaculture, it can start feeding after 35 h after insemination. We also found that, when rudiment and sessile pedicellariae appear in embryonic development, *A. crassispina* larvae is ready for metamorphosis at this stage. When the rudiment is found to be immature during the development of larvae, it will affect metamorphosis and lead to death. Therefore, it is very important to choose suitable bait. This study will be of great help in the large-scale aquaculture of *A. crassispina*.

**Author Contributions:** Conceptualization, Y.C., D.-S.D., W.-T.S. and C.-H.P.; methodology, Y.C.; software, W.-T.S.; validation, Y.C., D.-S.D. and W.-T.S.; formal analysis, D.-S.D.; investigation, D.-S.D. and Y.C.; resources, D.-S.D., C.G.S. and C.-H.P.; data curation, D.-S.D.; writing—original draft preparation, D.-S.D.; writing—review and editing, D.-S.D.; visualization, D.-S.D.; supervision, Y.C., C.G.S. and C.-H.P.; project administration, D.-S.D. All authors have read and agreed to the published version of the manuscript.

**Funding:** This research received no external funding.

**Institutional Review Board Statement:** No specific approval of research ethics committees was required to accomplish the goals of this study because experimental work was conducted with an unregulated invertebrate species.

**Informed Consent Statement:** Not applicable.

**Data Availability Statement:** Not applicable.

**Conflicts of Interest:** The authors declare no conflict of interest.

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
