# Peer review of "Effects of Marine Microalgae on the Developmental Growth of the Sea Urchin Larviculture Anthocidaris crassipina"

_fishes, doi:10.3390/fishes8060278_

Round 1

Reviewer 1 Report

This study was testing the initial bait for sea urchin under the captive condition. The experiment and outcome results are well done. However, my main concern in the survival rates through all the experimental groups. The ranges are 22 to 40 % and hence, these rates should be much less even they found optimal conditions. Therefore, even authors suggested C (20000) and C (30000) are optimal condition, the results would not be able to apply for future research. Why were survival rates so low for all experiments? Since there are so many similar or even same studies have been done, they need to describe and discuss for such lower survival rates. Furthermore, there should have various malformation when organisms were exposed unfavourable conditions. However, the authors did not state on it.

L21 Are C(2000) and C(3000) shall be C(20000) and C(30000)?

There are so many “Therefore” throughout the manuscript. In Introduction, they use five times. Need to rephrase.

In Introduction, the justification of this research is weak and hence, they need to strengthen it.

I recommend the authors send proofreading.

Author Response

Thanks to the reviewers for their suggestions, please refer to the PDF file.

Reviewer 2 Report

Please check for the below-mentioned issues throughout the manuscript and correct those. 

Check for correct spaces between the words or words and numbers. Better describe results with real treatment names, as using single characters would not provide a better idea. 

Select one, either protein or AA.

Check for correctly writing references within the text. Check the author's guidelines.

Need for a revised objective as no clear and strong sense is given.   

Check for the correct way of writing scientific names. Check throughout. 

Check for the correct SI units. 

Check for the correct way of writing the "p" value. 

Place the figures and tables immediately below they appear in the text. 

Thank you. 

Check for minor issues with the language.

Author Response

(The authors gave the same response as above.)

Reviewer 3 Report

The manuscript analyzes the effects of three different microalgae (Chaetoceros muelleri, Isochrysis galbana, and Dunaliella salina) to improve the survival and growth rates of the sea urchin larvae Anthocidaris crassipina.

The focal point for me it is the control group as it is not clear at all the choose of this group. On line 128-129, as reported “The no feeding group served as the control group (N)” what does it means? What kind of food (i.e., microalgae) did you use for the control group? It is not possible to growth sea urchin larvae for 12 days without microalgae, this cannot be considered a control group. Please add further explanation for the control group, i.e., how control larvae were cultured. In connection with this point, on Table 2 “The final rudiment length and final survival rate of sea urchin larvae after feeding for 12 days”, for the group N (the control group) it was reported a Final rudiment length (µm) = zero and a Final survival Rate = 0, what does it means? Again, in this direction, In Figure 7 (Differences in the developmental morphology of A. crassipina larvae fed different microalgae), images showing control larvae should be added.

Regarding the Materials and Methods section, on line 81-82, Authors reported that “the collected eggs and sperm were separated and stored in a 1000 mL beaker” Did you collect sperm directly in seawater? Generally, following KCL injection and in case of male, the standard procedure is to collect sperm dry and kept at 4°C until use (seawater determines sperm activation). An aliquot of sperm can be diluted to evaluate sperm quality and eventually to count it before fertilization. In case of eggs, after KCl injection and collection two or three washes using filtered or sterile sea water are needed to remove KCl. Please add further details and explanation on these specific points of sperm and eggs collection.

Other minor observations:

Line 69, A. crassipina should be written in italics “A. crassipina” check in all the manuscript, many other species are not reported in italics.

Line 216, please correct “grasprulas”

In the legend of Figure 5 it is reported a 200 µm scale bar that it is not present in the pictures, please add it.

In figure 5J there are 2 arrowheads that are not described

In table 2, what does “ppm” mean?

Author Response

(The authors gave the same response as above.)

Reviewer 4 Report

The paper titled “Effects of Marine Microalgae on the Developmental Growth of the Sea Urchin larviculture Anthocidaris crassipina”, reports important information related to the effects induced by microalgae in sea urchin. Several parameters were evaluated in order to study the development and growth of sea urchins. This approach is very interesting given the position of the sea urchin in the food chain and its important ecological role.

I suggest minor revisions

Figure 2. bar should be reported

Figure 5. also in this case a bar should be reported. In addition I think that Prism and 2-arm pluteus were not shown to scale of the fertilized egg.

Finally I suggest to the authors to expand the conclusions that must not only contain a summary of the results but also a relationship with what is added to the information already present in the literature.

Author Response

(The authors gave the same response as above.)

Round 2

Reviewer 1 Report

I understand the current study is "preliminary report" as you said. However, as a scientific paper, the outcomes need to show some positive data sets. I believe the high mortality rates found in this research should not become a reference. I strongly recommend the authors conduct the experiment again to find the real "optimal" condition with less mortality rate.

Extensive editing of English language is still required.

Author Response

Dear reviewer

Thanks to the reviewers, please refer to the PDF attachment file for comment reply.

Hope the revised manuscript will be published smoothly. Thank you.

De-sing, Ding

Reviewer 2 Report

Dear authors,

Thank you for submitting the revised version and addressing the issues raised. Please check again for any errors in the English language. But, it has improved to the level expected. 

The English language is satisfactory. 

Author Response

(The authors gave the same response as above.)

Reviewer 3 Report

The manuscript was improved following suggestions

Author Response

(The authors gave the same response as above.)
